# A General Description of Karst Types

**Márton Veress**

Department of Geography, Savaria University Centre, Eötvös Lóránd University, 9700 Szombathely, Hungary; veress.marton@sek.elte.hu

**Abstract:** This study includes a general description of the Earth's karst types based on literary data and field observations. An improved classification of karst types distinguishes the main group, group, and subgroup; and, a division of karst types involves a main karst type, karst type, subtype, variety, and non-individual karst type. The relation between karst type and karst area is described. The role of various characteristics of karsts in the development of primary, secondary, and tertiary karst types is analyzed. Their structure is studied, which includes a geomorphic agent, process, feature, feature assemblage, karst system and the characteristics of the bearing karst area. Dominant, tributary, and accessory features are distinguished. The conditions of the stability and the development of types are studied, transformation ways are classified, and the effect of climate on types is described.

**Keywords:** karst type; azonal karst type; zonal karst type; dynamic karst type; structure of karst type

## 1. Introduction

This study gives a general description of karst types. Thus, the hierarchical classification and structure of karst types, the characteristics of their stability, their development and transformation, and the relationship between the climate and karst types are overviewed.

The significance of geodiversity [1–4], to which karst also contributes, is given by the fact that it is the basis of the survival of biodiversity [5]. Karstifying rocks are widespread on the Earth (Figure 1). Karst areas can be put into karst types. The analysis of karst types enables better knowledge of karst diversity. However, the general description of karst types may also be an effective tool for a global overview of the karstic and the non-karstic processes working in karst areas. The concept of a karst type is based on a characteristic feature of karsts that some characteristics of certain karst areas and thus their karstification show similarities, while they can be significantly separated from other karst areas regarding other characteristics.

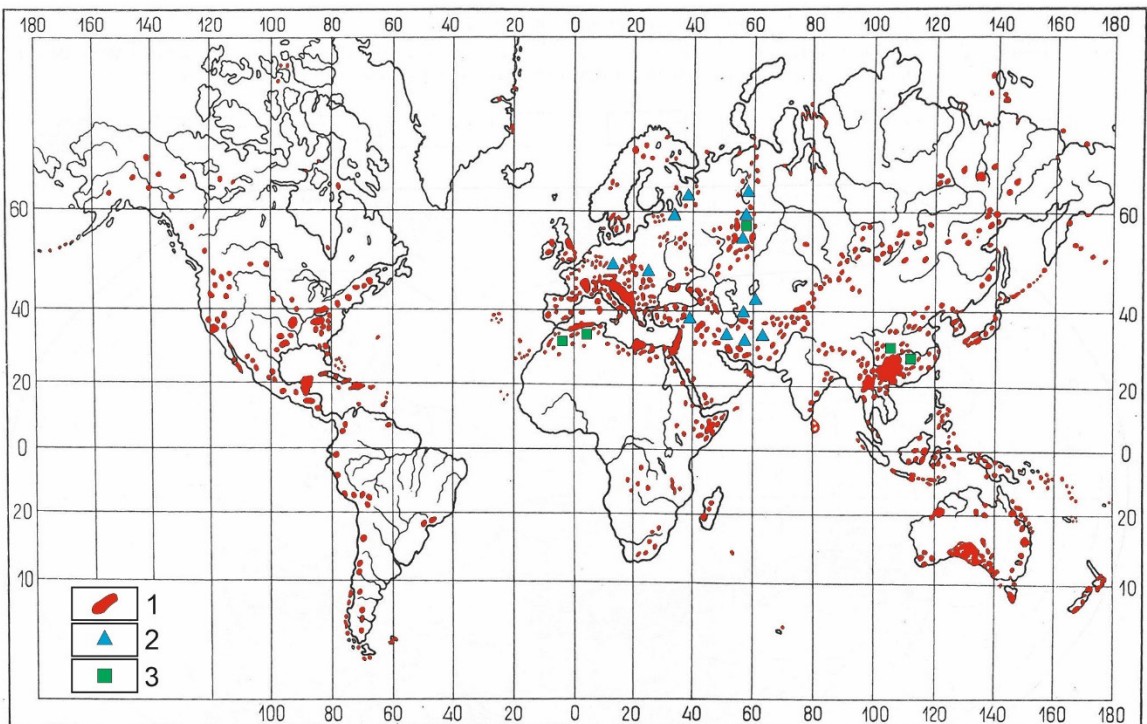

**Figure 1.** The Earth's karsts [6]. Legend: 1. limestone karst, 2. gypsum karst, 3. halite karst.

During research, various karst types were distinguished and described [7–16]; their comprehensive classification was made by Veress [17]. When karst areas are classified as karst types, the following are considered: altitude of karst, its geological characteristics (rock, structure), climate, hydrology, morphology, and geomorphic evolution. In some cases, various type names do not refer to other karsts (karstic content), but the different characteristic features of the karst area resulted in manifold terminology. Classification into types is based on some characteristics of the karst area, while putting the same karst area into another type is made according to other characteristics. Thus, the type names of geosynclinal karst and high mountain karst (mountain glaciokarst as well) refer to the same karst areas. The denudation of karst rocks happens by the process during which the rock gets into solution. At geosynclinal type, karst structure, at high-mountain type, climate-vegetation zonality and soil zonality, while at glaciokarst, the existence of glacial surface development are the viewpoints that determine classification. The dissolved material precipitates, while being transported to different distances. The dissolution intensity depends on rock characteristics (rock structure like fractures and faults are the most important discharge sites of the karst, but dissolution also takes place along them). However, it also depends on rock quality (evaporates are dissolved more intensively than carbonate rocks) and on the $CO_2$ content of water on carbonate rocks [12]. Since the $CO_2$ content decreases farther from the Equator through the intensity of soil life, the diversity, size, and density of karst features decreases farther from the Equator (and from the sea level) [18].

On karst, a spatially developed hydrology develops, but its surface is poor in streams because of the infiltration of meteoric water. The infiltrated water fills the gaps and cavities, created by itself, and its surface is the karstwater level. Its elevation is determined by the altitude of the surrounding terrain and it constitutes the base level of erosion of the karst together with it.

Karst areas are characterized by cavity formation, but the landscape is specific: surface features are mostly arheic. (It should be noted that if the karst is covered, features different from uncovered karst also develop in its area.) Arheic features are for example ponors, dolines, and poljes, while open features are karren and intermountain plains [14]. However,

among karst features can be mentioned remnant features such as inselbergs or the remnants of cave ceilings, the arches [19].

All areas built up of soluble rocks can be regarded as karst. These rocks are limestone, dolomite, gypsum, anhydrite, halite, calcareous conglomerates, and marl. These are dissolved by carbonated water and other acids (for example sulfuric acid, which is released during dissolution of rock with pyrite content) or only by distilled water. However, alkaline water can also have a dissolution effect on sandstones and conglomerates.

Because of their water storage capacity, karst areas have a significant role in drinking water supply, especially in areas with a dry climate. Paleokarst features have economic significance. They are sediment raps of minerals like bauxite, manganese, etc. [20,21].

Caves develop during underground, mainly dissolution, processes. These caves are significantly different in size, particularly on limestone karsts. They are also genetically diverse thus there are of ponor caves, spring caves, through caves, foot caves, etc. [12,19,22]. They often have an important role in the development of karst types (hypogene karst). In other cases, caves participate in karst type formation together with surface karst.

In addition to dissolution, non-karstic effects also take place on the karst for a shorter and a longer time. The type of effects is climate dependent. Thus, on temperate and cold-climate karsts, frost weathering, mass movements, and fluvial erosion are also widespread. The latter occurs if the karst is covered or the karstwater level is at the valley floor. In areas above the snow limit (the altitude of which depends on the distance from the Equator and on the Earth's global climate) glacial erosion is the main geomorphic agent. After ice regression too, glaciokarsts develop.

## 2. Classification of Karst Types

The areas of soluble rocks of the Earth (Figure 1) can be classified as karst types taking into consideration their karstic characteristics. (These involve the way and the intensity of karstification, the resulting karstic landscape, which can primarily be given by features.) These are postgenetic karsts at which karstification was preceded by rock formation. This study focuses on karst types representing such karsts. In addition to postgenetic karsts, syngenetic karsts can also be distinguished at which rock formation and karstification take place simultaneously or the two processes may alternate and thus happen repeatedly [23–25]. These are mostly characteristics of the intertidal zone.

The karst type usually covers a large area. Its features (maybe processes) are more or less different from other karst types since their characteristics or the affecting processes as well as their intensity and duration are different. Thus, in case of classification based on climate, the tropical belt is characterized by inselberg karst, while dolines are specific to the temperate karst. Karst types can be classified as units of various order, which constitute a hierarchical system [17]. Improving this classification, the units are as follows: main group, group, subgroup, and type. Two main groups can be distinguished: the karstic and the pseudokarstic main groups (Table 1).

The karstic postgenetic main group includes the karst types of karstifying areas on which dissolution takes place, while there is no dissolution (or dissolution is only subordinate) on the areas of the pseudokarstic group, but the features are karst-like [14] thus, for example, lava cave areas. In addition, various features (for example cavities) may develop on calciferous rocks during partial dissolution. The features of these rocks which developed during dissolution are also regarded as pseudokartic [26] and thus they can be put into the pseudokarstic main group. Such features occur on metamorphic rocks for example in the Hungarian Kőszeg Mountains [27]. Anthropogenic activities (for example mining) often also result in karst-like features, when anthropogenic cavity formation induces natural processes [28,29] or artificial water level lowering [30–33] results in the development of karst-like features (doline development). However, non-karstic natural effects (earthquake) may also trigger cover cavity collapse and thus doline development (though at this time the development of cover cavities is preceded by material transport into the karstic cavities of the bedrock). There is no sharp boundary between the criteria of

classification into the two main groups. Thus, the features of loess terrains partly develop by dissolution and by other processes (for example partly or exclusively by suffosion). Pseudokarsts are often of syngenetic karst character (for example, in case of lava caves, rock formation, and cavity formation take place simultaneously).

**Table 1.** Global, hierarchical system of karst types.

| Main Group | Characteristic Feature | Group | Characteristic Feature | Subgroup | Characteristic Feature |
|---|---|---|---|---|---|
| sygenetic karst | dissolution is simultaneous with rock development | - | - | - | - |
| postgenetic karst | dissolution takes place later than rock development | static | without landscape evolution | zonal | in case of climatic effect |
| | | | | azonal | without climatic effect |
| | | dynamic | considering geomorphic evolution | - | - |
| pseudokarst | another natural process e.g., lava flood | - | - | - | - |
| | partly anthropogenic effect | - | - | - | - |

Notice: Based on the data of [17].

The types of karstic main groups can also be classified as static and dynamic groups. At the static group, classification is possible by taking into consideration the actual state and characteristics of the karst area, while the dynamic group is based on the state of landscape evolution and thus to what extent the surface of a given karst approached the state of a karstic peneplain. The areas of the static karst types are in a certain stage of geomorphic evolution; thus, they represent a phase of the landscape evolution of the dynamic group.

Classification of dynamic karsts is made by distinguishing development phases occurring until the karstic peneplain state. The classification, the phases, and their characteristics may be different according to various authors [7,8,34], but probably karstic peneplanation may actually be different based on the type of the given karst area. This is well revealed for example by the landscape evolution of tropical karsts [34] or of covered karsts [17,35] since geomorphic evolution depends on the initial state of the karst area and on the hydrological, geological, and altitudinal characteristics and climate of the karst too. According to this, static karsts can be put into different types considering their dynamics based on the state of the surface. For example, on Mediterranean karst, according to Grund [7] carbonate karsts can be young karsts (there are dolines) and old karsts (there are karst hills), while tropical karsts can be of fengcong, fenglin, and gufeng types [15,16], or according to Waltham and Fookes [34], they can be classified as juvenile karst, youthful karst, mature karst, complex karst, and extreme karst types.

Within the static group, azonal and zonal subgroups can be distinguished. Regarding another approach, according to Balázs [6], karstification and thus karst occurs at sites where both azonal and zonal conditions are present. Classification of karst areas as azonal karst types and thus the determination of the type is possible based on non-climatic factors (karstification age; rock; rock development; the expansion, altitude, and surface of the karst; the temperature of karstwater; the origin of karstwater, etc.). Classification into the zonal subgroup happens with the consideration of the climate of the karst area. Surface features of zonal and azonal karsts are specific (Figures 2–4). The same karst area can also be put both into the azonal and zonal subgroup and into the dynamic group as well. However, azonal karst types (and thus karst areas) can occur under any climate, though

their processes and features may be different depending on their occurrence under various climates. Since their climate is the same, the karst areas of some zonal karst types have only the same kind of features, but the size and the density of the features may, be different depending on the characteristics of the karst and on the age of karstification for example, on temperate karsts. This is not necessarily true for tropical karsts, where the feature diversity is great. However, there may occur fengcong and fenglin (depending on the maturity of landscape evolution) on inselberg karst (the two types are different regarding the shape and the density of mountains). At the same time, temperate features (e.g., dolines) are also present.

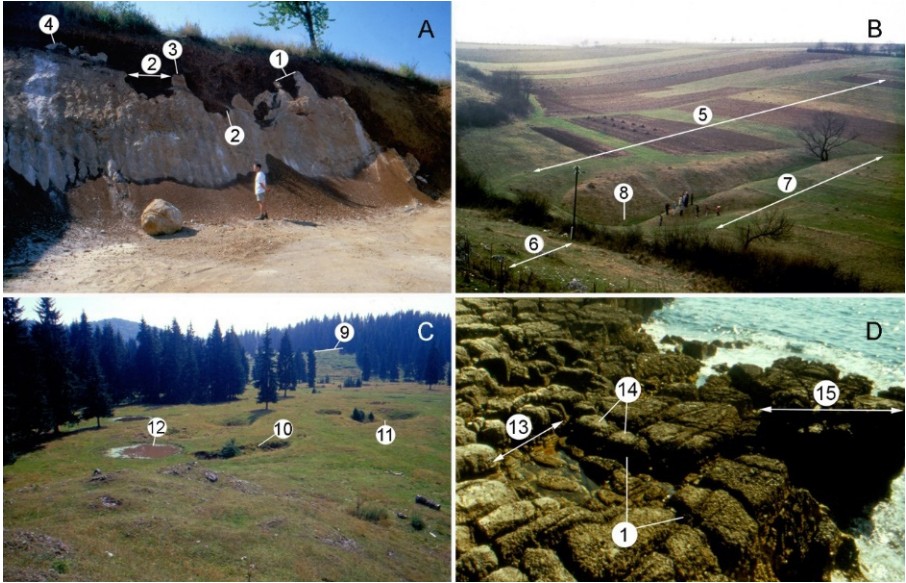

**Figure 2.** Azonal karsts according to rock development and altitude (photograph taken by Márton Veress): (**A**) soil-covered karst (Croatia), (**B**) mixed allogenic–autogenic karst (Aggtelek Karst, Hungary), (**C**) covered karst (Pádis, Romania), (**D**) coastal karst (Locrum, Croatia). Legend: 1. grike, 2. kamenitza-like depression on the floor, 3. ridge between kamenitza, 4. detached section of bedrock, 5. covered karst, 6. bare karst, 7. blind valley, 8. ponor, 9. margin of depression, 10. elongated subsidence doline, 11. uvala like subsidence doline, 12. inactive subsidence doline, 13. kamenitza, 14. spitzkarren, 15. phytokarst.

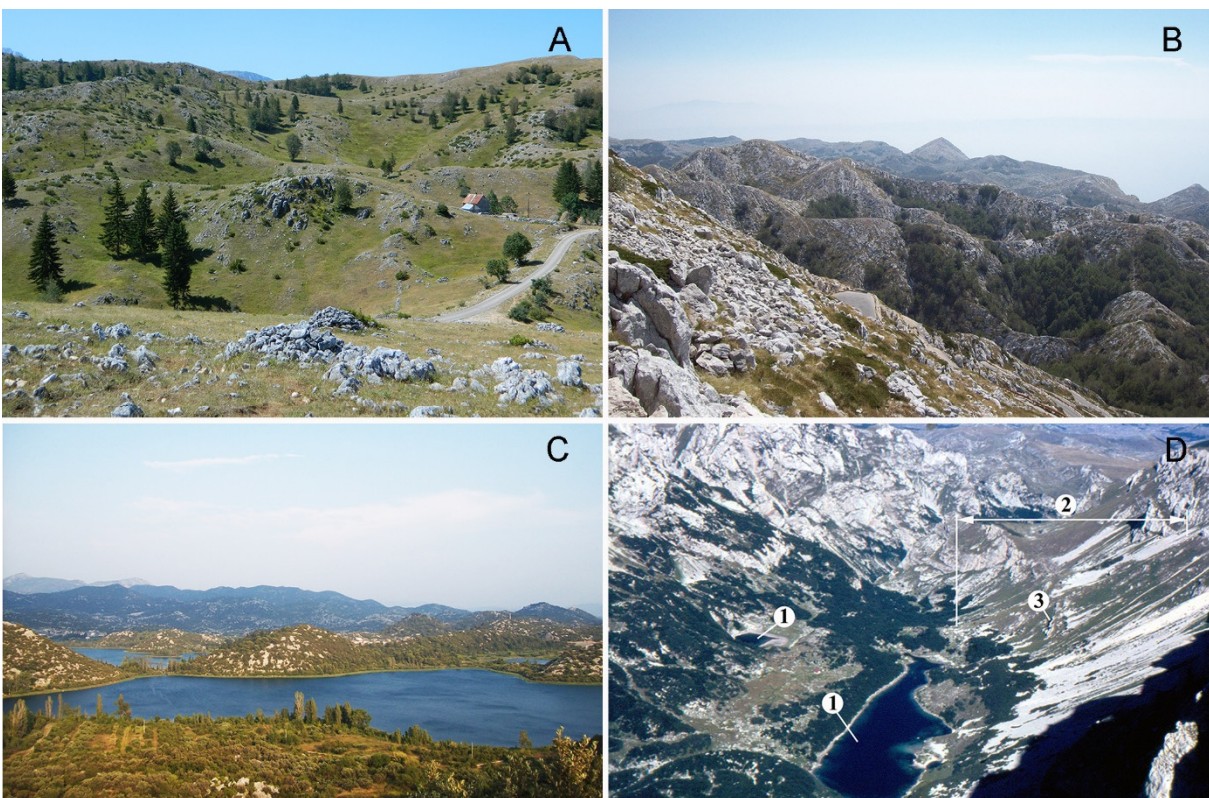

**Figure 3.** Azonal karsts according to morphology and effects (photograph taken by Márton Veress): (**A**) karst with dolines (Durmitor, Montenegro), (**B**) polygonal karst (Biokovo, Croatia), (**C**) karst with karst hills (Croatia), (**D**) glaciokarst (Durmitor). Legend: 1. rock basin, 2. debris slope, 3. margin of partial doline covered with debris.

The zonal karst type is not an element of the azonal karst type, but the azonal is an element of the zonal. In addition, a karst type belonging to a given azonal subgroup may also be present on several zonal (but azonal) karst type (for example the mixed allogenic–autogenic karst may occur on temperate karst but on tropical karst, too). As already mentioned, karsts belonging to the azonal subgroup are made up of the same feature variations and feature assemblages (except if the age of karstification is different). The features and feature assemblages of the karsts of karst types belonging to the azonal subgroup are controlled by climate. Their size, density, and pattern may be different. At feature assemblages, the proportion of certain features may alternate, there may be a lack of some features (for example subsidence dolines), mainly those that are not climate dependent features such as the above-mentioned dolines.

A group of azonal karsts can be distinguished, which were only dependent on climatic effect earlier (for example paleokarst) and another group whose members do not depend on climate at present either. These are the types, in the development of which only the flow system of the karst plays a role (thermal karst, eogenetic karst, telogenetic karst, certain pseudokarsts, for example, lava caves). In case of these, the possible appearance of landscape features and the degree of their distribution may depend on the degree of burial and exhumation (paleokarst). In case of thermal karsts, surface and subsurface features are foreign and genetically independent of each other, surface features may be rudimentary, or they do not develop at all. In some parts of the Transdanubian Mountains (Hungary), surface features are only represented by weakly-developed covered karst features, while independently of this, subsurface features are well-developed (hydrothermal caves are specific for example in the Hungarian Buda Mountains).

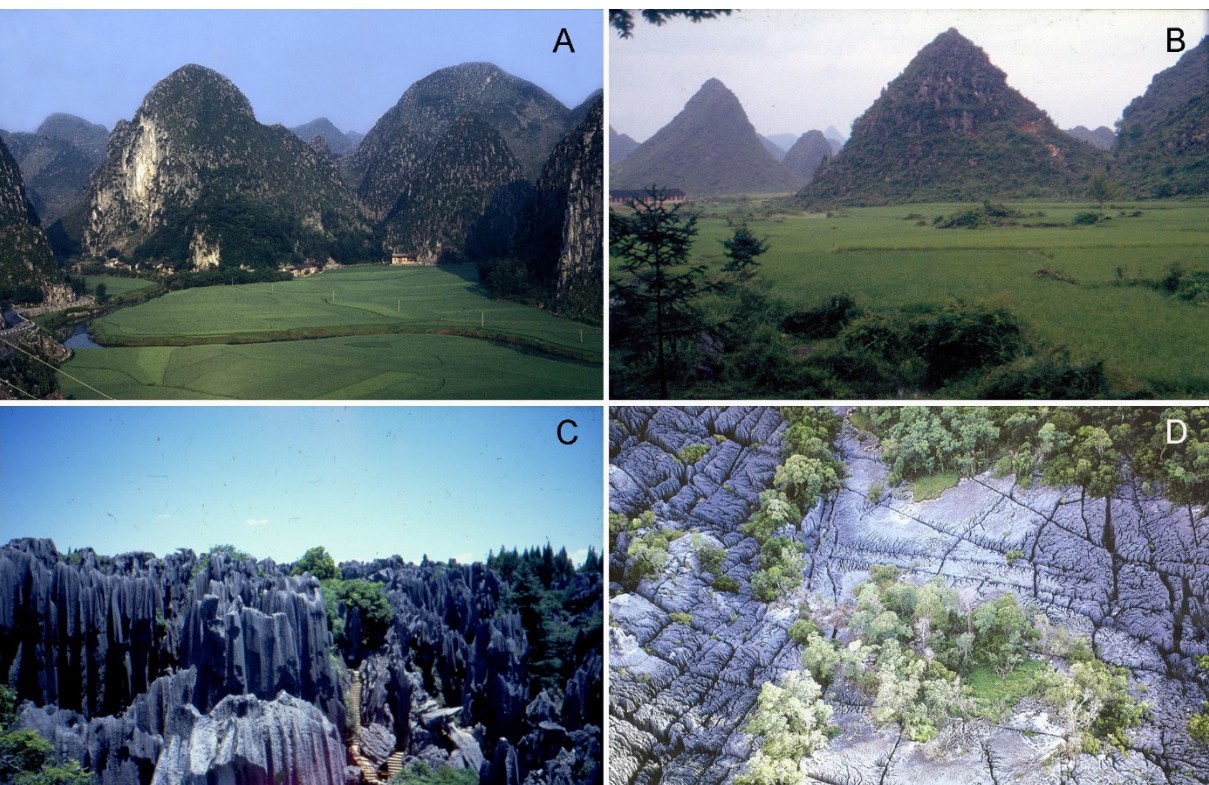

**Figure 4.** Zonal (tropical) karsts: (**A**) fengcong (China), (**B**) fenglin (China), (**C**) stone forest karst (Lunan, China) (photographs (**A**–**C**) taken by Márton Veress), (**D**) tsingy (Bemaraha tsingy, Madagascar) [36].

## 3. Karst Type and Karst Area

A karst area can be of homogeneous type thus, belonging to only one karst type or heterogeneous type. Homogenous karst is rarer such as the subtypes of tropical karren. Different parts of heterogeneous karsts belong to various karst types (for example the Dinarides). In this case, altitude, rock quality, and morphology control the classification. Belonging to various types can be regular such as for example that related to rock quality. Here, mixed allogenic–autogenic karst is aligned along non-karstic rocks. It may also be irregular (accidental), for example in high mountains, the site of soil-covered karst and bare karst and their position as compared to each other is determined by valleys and ridges (summits) [37].

The expansion of the karst type may spread beyond the boundary of a given karst area. In this case, the karst type involves several karst areas (for example, South China). These are mostly the zonal karsts.

A karst area always belongs to only one zonal and dynamic type at the same time. However, the same karst can always be put into several azonal types (except evaporate karsts), but usually only one type is dominant. For example, the Totes Gebirge (Austria) is carbonate karst according to rock, geosynclinals karst according to structure, high mountain karst according to altitude, glaciokarst according to process, and soil-covered (partly bare) karst according to coveredness. Based on the image of the mountains, the glaciokarst type is the dominant (Figure 3D). The Buda Mountains is medium-mountain karst according to their altitude, they are soil-covered karst according to their cover, horst type karst according to their structure, and thermal karst according to their hydrology. According to their predominant type, they are thermal karst.

## 4. Structure and Order of Karst Type

Elements of the structure of karst types are features (feature assemblages), karst systems, geomorphic agents, processes, and the characteristics of the areas of the type. However, not every karst type has features (for example certain paleokarsts or telogenetic karst).

The characteristics of karst areas are the local manifestations of the Earth's belts (crust, atmosphere, water, soil, and biosphere). The characteristics (properties) of karst areas, which may show significant variations regarding karst areas, are rock quality, crust structure, evolution, vertical movements (their duration, degree, and velocity), actual altitude, position compared to the base level of erosion, changes of the base level of erosion, solubility, weathering residue and its characteristics, the presence or lack of soil and biosphere, the characteristics of the surface, precipitation (its quantity and distribution), temperature, and the characteristics of the karst surface. The characteristics of the karst areas regulate and give a frame for geomorphic agents. Thus, vertical movements control geomorphic agents and their intensity through climate, but they also affect the position of the karstwater level as compared to the surface, which may significantly control the karstification of the area. The characteristics of the karst area collectively affect and shape the type of karst at a given site. The effect of certain characteristics is different in the development of the karst type. Thus, the effect of precipitation is much more important than for example of crust structure.

Processes affecting the karst surface are feature formation and the resulting denudation or accumulation and reworking. A separate group is represented by hydrological processes such as water inflows and outflows, karstwater phenomena (water level fluctuation, karstwater flow, the relation of karstwater and groundwater as compared to each other). Some processes are not specific (feature development), others are karst type specific (for example hydrological processes). However, the quantitative characteristics of the latter are also specific according to karst area (for example the degree of infiltration in case of the latter).

Agents forming the karst surface may be outer (dissolution, fluvial erosion, and glacial erosion, biosphere, root system, snow erosion, chemical weathering, mass movements, pluvial erosion, insolation weathering, and frost weathering) and inner (the internal heat of the Earth, earthquakes). Some agents are mostly not type specific (dissolution), but type specific agents (frost weathering, glacial erosion) and area specific agents (the internal heat of the Earth) also occur among them. Geomorphic agents affecting a karst type may be permanent (dissolution) or intermittent (erosion). Primarily, karst processes (dissolution, concretion) affect, but non-karstic processes (for example frost weathering) also occur. The quality, number, and degree of non-karstic geomorphic agents depend on climate (on the distance from the Equator), on the characteristics of the karst (on its coveredness, on the presence of non-karstic rocks, and on the position of the karstwater level).

On the surface of the karst, large features, medium features, and small features can be distinguished. The size of large features exceeds several hundreds of meters, that of medium features is between some metres and some hundred meters, the width and depth of small features (karren) is below some metres, but the size of megakarren [38,39] is larger than several tens of metres. The features may be simple (when constituted by one feature only) and complex (made up of several similar features), homogeneous and heterogeneous. There is no other feature on (in) the homogeneous feature, but there is on (in) the heterogeneous feature. Mostly, large and medium features can be heterogeneous. On large features, medium and small features may occur (for example on inselbergs and between them, dolines and karren), while on medium features, small features (karren in dolines) may be found. The features may be complex genetically, too. In this case, several effects influence feature development. (For example, dissolution and collapse at dropout dolines.) Heterogeneous features are also complex genetically. For example, regarding the development of poljes they are complex as a matter of course [11,40], but features of various shape and genetics (karst hill, subsidence doline) may also occur on their floor.

A karst type mostly has features and a combination of features that is only specific to that certain karst type because of the impacts affecting the karsts and of their characteristics. However, there are features specific of several karst types (doline) and features characteristic of all karst types (cave). These are cosmopolite features.

The features of karst types may be dominant features, tributary features, and accessory features (Table 2). This classification is mainly valid for surface features. The cosmopolite character is dominant at subsurface features since few cave types are characteristic at the beginning of karstification. However, caves may become differentiated during the development of the karst and only some cave types will be more and more dominant, or they may develop on a karst type to the effect of surface impacts and geomorphic evolution taking place there. For example, on mixed allogenic–autogenic karst which develop during suitable karst development as a matter of course, erosion caves (inflow caves and then through caves, may be storeyed caves) develop.

**Table 2.** Features and feature assemblages of various karst types.

| Type | Group, Subgroup | Aspect of Karst Classification | Dominant Feature | Tributary Feature | Accessory Feature | Feature Assemblage |
|---|---|---|---|---|---|---|
| complex karst | dynamic karst | tropical karst (Waltham, Fookes 2003) | inselberg, remnant hills, solution-buried dolines, caves | soil karren, subsidence dolines | bare karren | karst mountains and dolines (G) |
| halite karst | azonal karst | rock | bare karren, dolines, breccia pipe | concretion | - | breccia pipe and dolines (G) |
| warm-water karst | azonal karst | karst water temperature | hydrothermal cave, hydrothermal minerals | cold water cave | doline, concretion | hydrothermal features (E) |
| plateau karst | azonal karst | expansion | solution doline, uvala | soil karren, concretions | mound (hills) | doline system and uvala (G) |
| concealed karst (covered karst) | azonal karst | sediment | subsidence doline, shaft, | - | soil karren | subsidence doline and shaft (G) |
| fenglin | zonal karst | climate | isolated tower, intermountain plain | different dolines, karst hills, shafts, pinnacles, bare karren, soil karren, concretions, active and non-active caves | - | karstic mountain and plain (G) |
| glaciokarst | azonal karst | effect | karren (dominant bare karren), shaft cave | schachtdoline | subsidence doline | karren system (E) |
| mixed allogenic–autogenic karst | azonal karst | origin of water | bind valley, ponor, erosion cave (inflow cave and through cave) | - | - | surface and subsurface erosion systems(G) |

Notice: G: genetic feature assemblage, E: environmental feature assemblage.

The majority of surface features is climate dependent (solution doline), but there may be features partly dependent on climate (suffosion doline) or completely independent of climate (collapse doline). (Caves are also directly independent of climate.) Predominant features are always present, they determine the image of the type in the karst areas of a given karst type. The predominant feature is a leading feature if it is present on only one karst type (for example the tower hill on fenglin karst). The features of karst types may be unique. Only some or a few features are in the karst areas on a type. Their appearance is unique as their development is accidental (for example at collapse dolines, the collapse of the cave ceiling) or several factors are necessary for their development (for example gorges). Tributary features are less widespread, but they are present (for example on doline karst or on autogenic karst, the dry valley). Accessory features are not necessarily present, in case of their lack, the karst area still belongs to the same karst type (for example on doline karst,

the karst hill can be present or absent). Unique features may also be absent, but this does not modify the karst type.

Feature assemblages (Table 2) may be developmental feature assemblages (the participating features presuppose each other genetically) and environmental feature assemblages (without genetic relationship between them). The features of developmental feature assemblages develop from each other. The existence of a feature brings forth the development of another feature, for example, on fenglin karst, inselberg, and intermountain plain (the inselberg is formed during the development of the intermountain plain). However, subsurface features may also belong to them. For example, on mixed allogenic–autogenic karst, erosion cave types develop to the effect of surface features (blind valley, ponor). Environmental feature assemblages are features which occur in a given environment together. For example, on bare karsts, rillenkarren, rinnenkarren, and meanderkarren occur together.

Karst systems are the largest units of karst types, which are material flows (wet systems) and energy flows. Features and feature assemblages develop along flow (filtration) paths, which are the circulation systems of karsts. They go on the whole part of a given karst or on its geologically, hydrologically independent part. Thus, for example where there are impermeable beds above the main flow system on the karst locally, an independent flow system develops. Taking this into consideration, epigene and hypogene karst types are distinguished [41,42]. The features and feature assemblages of epigene karsts develop along local flow paths or along the descending (infiltrating) branches of regional flow paths. Feature assemblages of hypogene karsts are formed along the ascending flow path parts of regional systems. Here, several processes may play a role in the development of the features [41]. At epigene systems, surface karst features are predominant, while at the ascending branches of regional flows (hypogene system), subsurface features are dominant.

The characteristics of a karst area (the geological, climatic, and hydrological characteristics of the area as well as the properties of soil, biosphere, and karst surface) affect each other and the karst, too. The characteristics are of different levels based on the order of their interaction (Figure 5). Karst types may be primary, secondary, and tertiary depending on the level of the karst characteristics that take part in the development of the type (Figure 5). Some characteristics of the karst participate in the development of the types separately but together, too. The development of the primary type depends on geological conditions and climate. Such karst may be mostly salt karsts or some of them. (There is no primary type among carbonate karsts.) In addition to the above-mentioned factors, altitude, hydrology, and soil (biosphere) also play a role in the development of secondary types. All the already mentioned properties and the surface of the karst also have a role in the development of tertiary karsts.

Karst types can be classified according to their complexity. Thus, main karst type, karst type, subkarst type, variety (Table 3), and non-individual karst type can be distinguished. Not every karst type can be put into a main karst type. A specific main karst type is the tropical karst which has two karst types, the inselberg karst and the tropical karren (but temperate karsts have specific variations under terrestrial climate and oceanic climate as well). Karst types are more and more complex toward the Equator (Table 3).

Within the type, the subtype is a smaller unit and thus it covers a smaller part of a given karst area. Subtypes can be distinguished according to the specific features of karst areas or the features of the karst (Table 3). Thus, in case of classification based on characteristics, crust structure may be the basis of classification. Taking into consideration the varieties of the features, subtypes can be distinguished for example at tropical karren. Here, the shape and the size of the mounds of the subtypes (pinnacle) are different.

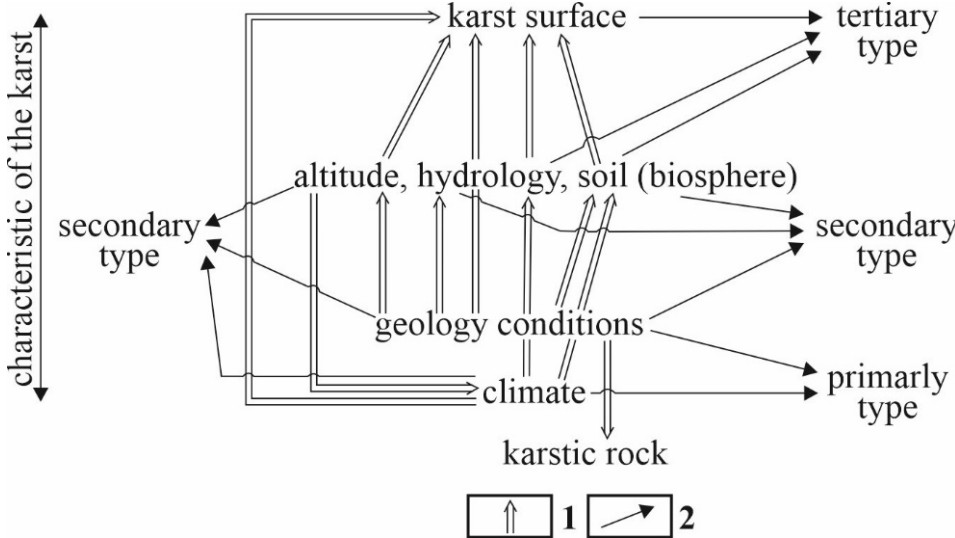

**Figure 5.** Interaction of factors determining karst types and their role in the development of karst types of various orders. Legend: 1. Interaction of the determining factors, 2. Karst type developed by a given determining factor.

**Table 3.** Some examples for the classification of karst types.

| Main Karst Type | Karst Type | Subtype | Variety |
|---|---|---|---|
| tropical karst | fengcong | fengcong plain [43] | - |
| | | fengcong depression [43] | - |
| | | fengcong canyon [43] | - |
| | fenglin | uplifted fengcong [16] | - |
| | | karst cones fenglin [43] | - |
| | | karst tower fenglin [43] | - |
| | | isolated tower fenglin [43] | - |
| | | uplifted fenglin [16] | - |
| | | rainforest karst [44] | - |
| | gufeng [16] | - | - |
| | karst archipelago | - | - |
| | tropical karren | stone forest | |
| | | arête | - |
| | | tsingy | pinnacle [45] blade [45] cling [45] |
| evaporites | gypsum karst | - | - |
| | salt karst | - | - |
| | glaciokarst | alpine [19] | Schichttrippenkarst, Schichttreppenkarst [46] |
| | | continental [19] | Schichttrippenkarst, Schichttreppenkarst [46] |
| | geosynclinal karst [13] | miogeosynclinal karst | - |
| | | eugeosynclinal karszt | - |

Varieties may belong to a type or to a subtype. At varieties, a feature of the bearing karst type will be predominant or exclusive. For example, karren are predominant or exclusive on the "Schichttreppenkarst" and "Schichttrippenkarst" varieties of glaciokarst [46] (Figure 6). The features may be of special patterns. Thus, for example on cockpit karst, which is a variety of fengcong, mounds enclose depressions. The non-individual karst type involves karst which may belong to any karst type (for example paleokarst).

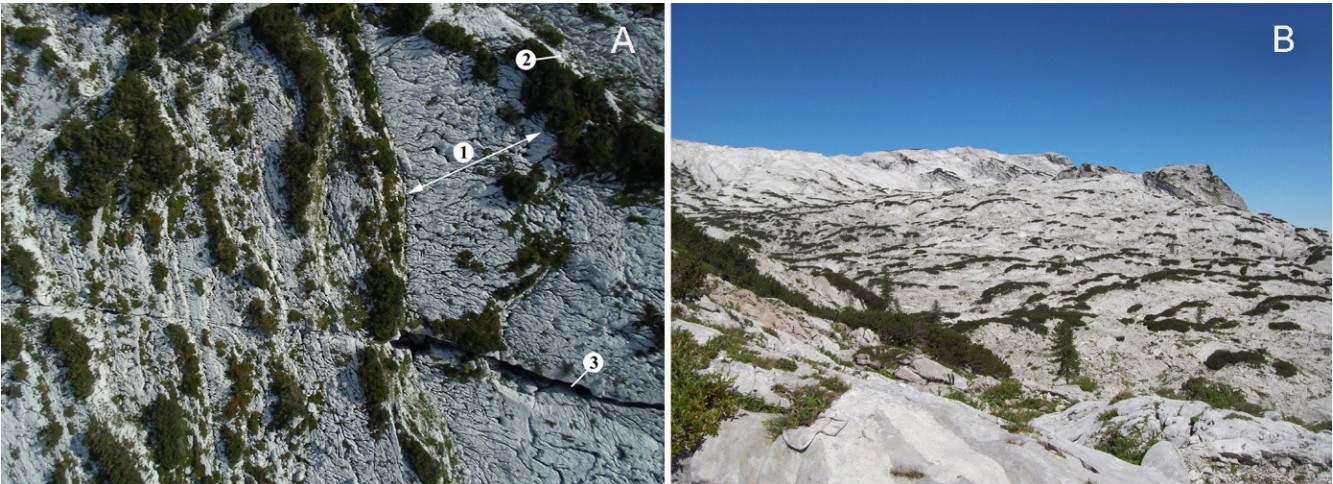

**Figure 6.** "Schichttreppenkarst" (**A**) and "Schichttrippenkarst" (**B**) according to Bögli [46], horizontal beds on A (aerial photograph), inclined beds on B (Totes Gebirge, Austria). Legend: 1. bedding plane with rinnenkarren, 2. scarp front, 3. giant grike. (Photographs taken by Márton Veress).

## 5. Stability and Development of Karst Types

The state of a karst area and thus of a karst type can be stable and unstable. If the karst area is unstable, it changes karst type. Features may become filled, truncated, and transformed. During their transformation they often coalesce and take part in the construction of features belonging to the new type. The transformation, which may be of very different rate, is reversible or irreversible. If it is reversible, two (or more) karst types change each other repeatedly (Figure 7). If it is irreversible, the transformation is of one-way (Figure 8). The change of karst type occurs because its state of equilibrium with its environment ceases (for example climate change or covering occurs) or because the geomorphic agents affecting the karst area significantly change its characteristic features. For example, the non-karstic cover is denuded. After all, these phenomena can also be traced back to the change of the equilibrium state. The chance of the permanence of stable state increases and thus the chance of karst transformation decreases if other effects are unable to modify karstification. The chance of this is great if the karst is bare or soil-covered and more elevated than its environment. In this case non-karstic processes regress, the change of karst type is only controlled by the development of its surface, and the karst reaches its stable state by getting into the state of karstic peneplain.

The stable state can survive even in case of the broad change of regulating conditions. For example, on the already mentioned covered karst, the partial denudation of superficial deposit (the duration of denudation may be longer or may even be interrupted) does not change the karst type.

The development of karst type is caused by karstic and non-karstic geomorphic agents and processes, which have or had a permanent effect in a karst area and which are controlled by the characteristics of the karst area. Since they have an effect in space (geomorphic agents and processes), the karst areas develop into karst types. Although geomorphic agents affect everywhere, their degree is influenced by the climatic environment, solubility, the actual state of the karst (elevation, surface), the geological state (rock quality, crust structure), and the hydrological state. The spatial pattern of geomorphic agents and processes influence

the arrangement of the Earth's karst areas into types and their pattern. The development of the karst type results in the development of a typical landscape. The landscape reacts upon the development and the transformation of the karst type.

Geomorphic agents do not have a similar role in the development of karst types. They may be primary, having a basic role in the development of the type (dissolution), and secondary with a subordinate role (erosion). Some agents take part in the development of every karst type (dissolution), while others participate only in the formation of one (glacial erosion) or some karst types (fluvial erosion).

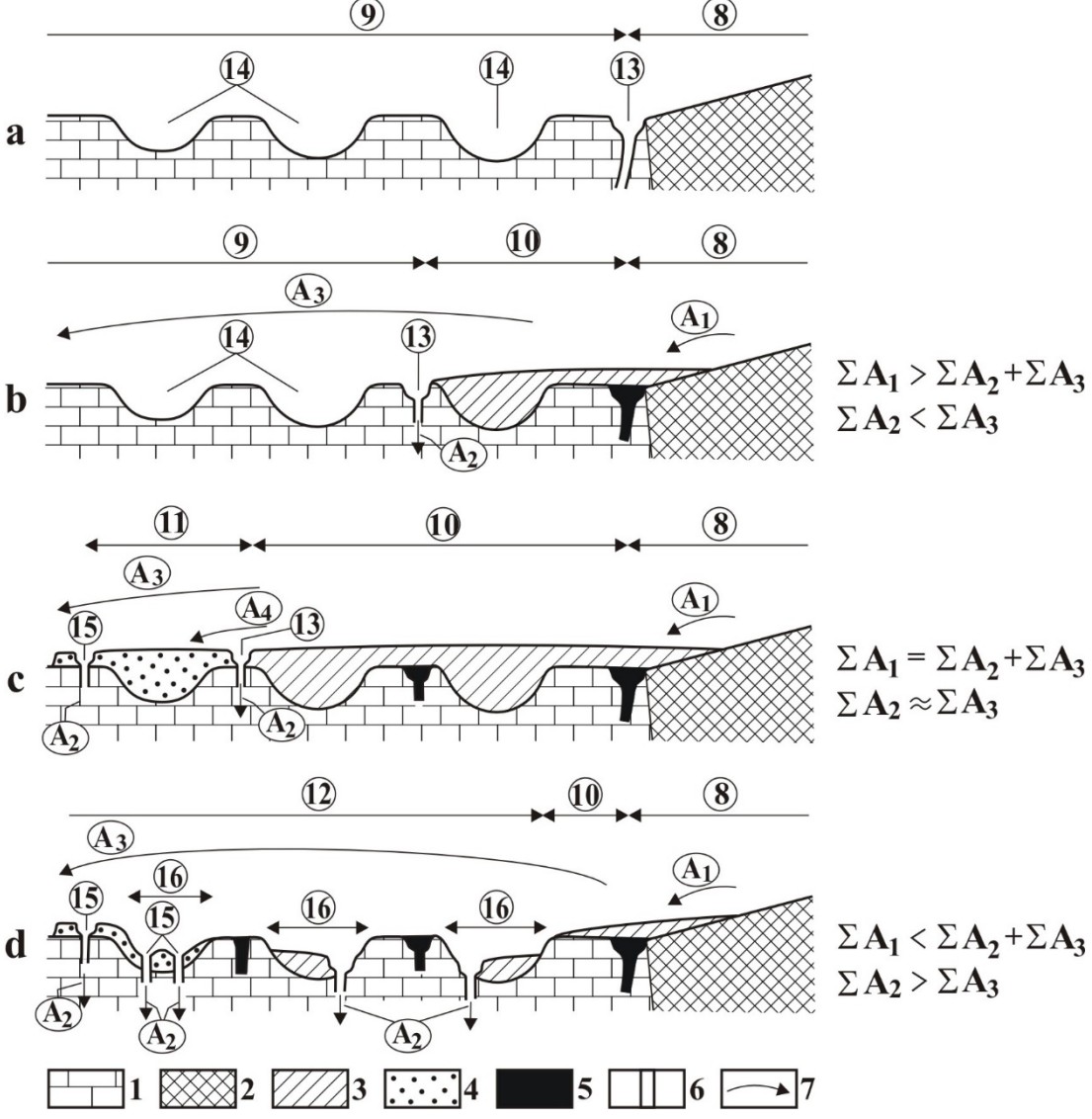

**Figure 7.** Reversible transformation on covered karst [35]. Legend: 1, limestone, 2, consolidated non-karstic rock (impermeable), 3, non-consolidated, impermeable, non-karstic rock, 4, non-impermeable, non-karstic rock, 5, ponor fill, 6, karst conduit, 7, sediment transport, 8, non-karstic terrain (buried karst), 9, bare karst, 10, cryptokarst zone, 11, concealed karst, 12, mixed composite karst, 13, ponor, 14, paleodoline, 15, subsidence doline, 16, true DSD. $A_1$ sediment transport upon the karst, $A_2$ sediment transport into the karst, $A_3$ outward sediment transport from the karst (fluvial), $A_4$ sediment redeposition within the covered karst, (**a**) bare karst, (**b**) beginning of covered karst evolution (the margin of the cover advances on the karst), (**c**) the bare karst transforms into covered karst (with cryptokarst and concealed karst zone), (**d**) the covered karst transforms into bare karst (the margin of the cover is retreating on the karst, the covered karst patches are shrinking).

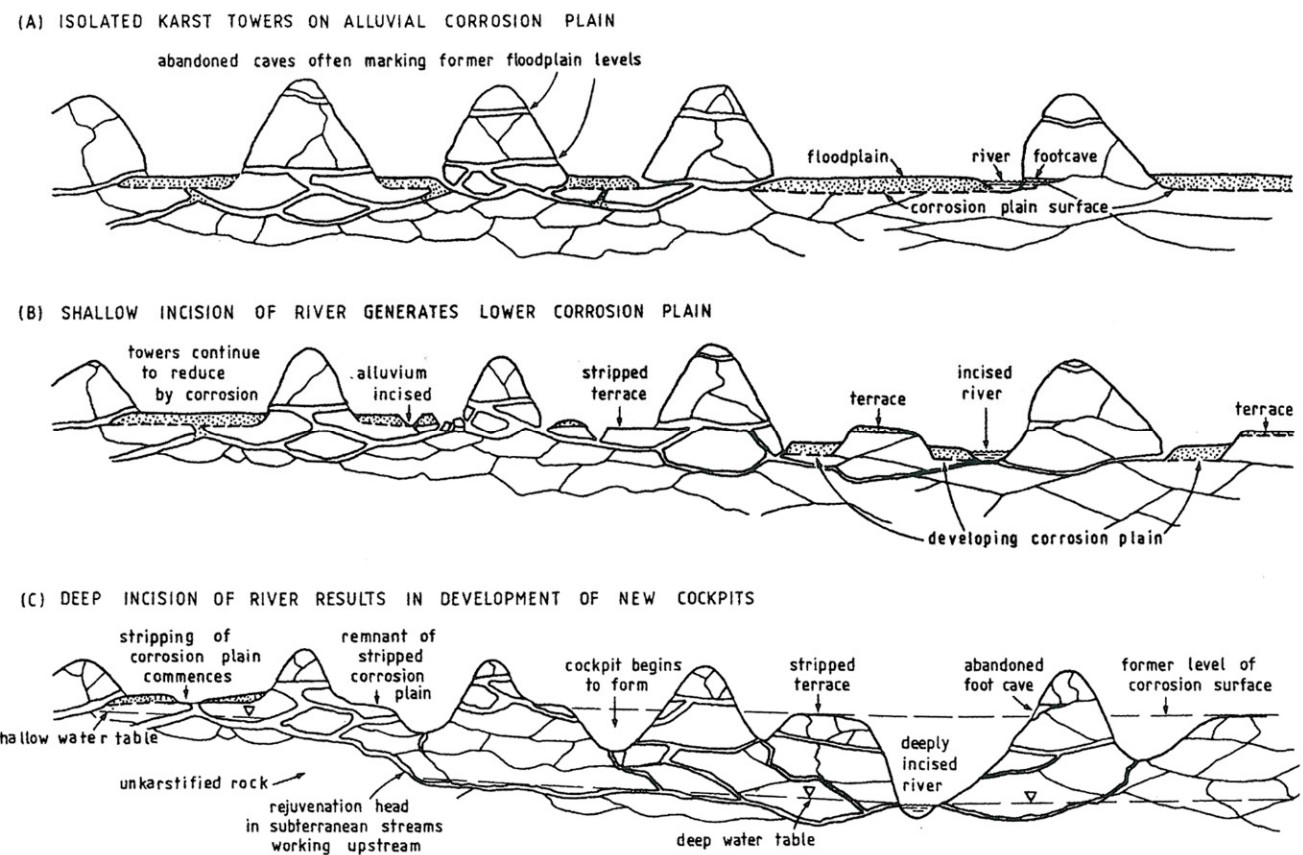

**Figure 8.** Irreversible transformation on tropical karst [19]. (**A**) (isolated karst towers on alluvial corrosion plain, (**B**) Shallow incision of river generates lower corrosion plain, (**C**) Deep incision of river results in development of new fengcong. Note that in (**C**) the isolated hills of the previous phase may become the tops of the highest tier of cones in the modern phase.

## 6. Climate and Karst Type

Climatic effect may be manifested on karst types in the following way.

- The features of azonal karsts also depend on climate. For example, on salt karst, features are absent in the case of a lot of precipitation since the salt mound is destroyed during intensive dissolution (if the surface does not rise intensively), but in the case of less precipitation, a diverse landscape develops.
- The effectiveness of climatic impact is determined by non-climatic factors (for example, in the temperate climatic zone, the features of the karst depend on whether the karst is of carbonate or evaporate material and for carbonate karst the characteristics of the rock).
- The diversity of karst features depends on climate. The closer a zonal karst type to the Equator (particularly if it is carbonate karst), the greater the diversity of its features, while the farther the karst type from the Equator, the smaller the diversity [18]. Dissolution intensity is larger and larger closer to the Equator since the quantity of biogenic $CO_2$ increases [12,47,48]. Therefore, the increase of diversity at large and medium features is caused by the increase of dissolution intensity [18]. The great diversity of small features in the tropical zone is enabled by bare slopes of various inclination since these types of slopes developed as a result of intensive karstification [18,49]. The degree of diversity also depends on altitude. With higher altitude, the degree of the diversity of large and medium features decreases, except karren features (small features) as a result of the decrease of dissolution intensity. At 1600–2100 m, the diversity of karren features increases because of bare slopes with diverse inclination [49]. The latter can be traced back to former glacial erosion.

- The degree of the heterogeneity of the features is also increasing toward the Equator. The change of the degree of heterogeneity is also related to dissolution intensity [50].
- In climate zones being closer to the Equator, climatic karsts are more diverse, too (Table 3). The number of the zonal karst types of the tropical zone exceeds the number of the zonal karsts of the temperate zone (Table 2).
- Climate change modifies the pattern of the Earth's karst types. A karst area may get into another zonal karst type.

## 7. Conclusions

A karst type may occupy a karst area, but its expansion can be smaller and larger, too. Parts of karst types are features, karst systems, geomorphic agents and processes, and karst hydrology. Karst features can be classified in several ways (e.g., size or complexity). Features may be dominant, tributary features, and accessory features. There is a genetic relationship between the features of feature assemblages, disregarding environmental feature assemblages. Karst systems are constituted by features belonging to given flow paths.

Karst types may be primary, secondary, tertiary, depending on the level of the effects participating in their formation. Karst type may be the main karst type, subtype, and non-individual karst type. According to their state, karst types may be stable or non-stable.

Climate determines the karst type through non-climatic factors. Climate affects the diversity and the heterogeneity of the features of karst types. Climate change may result in the transformation of karst types.

Geomorphic agents create impacts affecting the karst. Geomorphic agents and the properties of the karst determine the image of the karst, which is manifested in the type of the karst. The state (and existence) of the karst is given by the actual state and balance of geomorphic agents and the properties of the karst. This is reflected in the image of the karst area which can be described by the landscape with more or less accuracy.

Geomorphic agents, mostly dissolution, and their intensity affect the distribution and the expansion of zonal karst types. The complexity of zonal karsts increases by the growth of dissolution intensity thus toward the Equator. The properties of a karst area control the number, type, and characteristics of azonal karst types occurring on a zonal karst.

**Funding:** No funds, grants, or other support were received.

**Conflicts of Interest:** The author declares that he has no conflict of interest with regard to this article. The author declares they have no financial interest. The author has no financial or proprietary interest in any material discussed in this article.

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
