# Peer review of "A General Description of Karst Types"

_encyclopedia, doi:10.3390/encyclopedia2020073_

Round 1

Reviewer 1 Report

The situation with this paper is rather strange.  The data represented in the article of course are right and interesting.  But nearly the same possible to find in many sources including WIKIpedia and even in the handbooks for the students.  Some innovations are in the introduction of the miogeosynclinal karst - eugeosynclinal karszt.  Though this terminology was nearly reduced during last 40 years.

  The systematization of the types of the karst of course is not completely new but have some interesting moments concerning the types of climate and morphology, dynamics etc.

From the other point of view – there are three major chemical groups –1. carbonates including dolomites MgCO3 and   limestones and marbles MgCO3.  3.  Sulfites –gypsum and anhydrites 3. Evaporites – various salts.  It will be  more interesting also to have more  details about the  types and morphology of the two last groups  because the major descriptions are devoted to the typical limestones.

There are much more types.  For example I the Sothern Siberia some large caves are located within the limestone conglomerates with the essential part clay material in the cement.  So the caves contains huge amount of the clays.  Also the marls have there own features.

So more detail  characteristics and dependence between chemistry and morphology are highly wishible.

The so called ore Karsts  processes taking place near the sulfide  deposits will be interesting to mention.  In such processes the aggressive role of the H2SO4 () is very important.

The paper includes very god and interesting photos of the karst landscapes.  But the karst  forms in the earth interior will be interesting.     

It will be also interesting also to say several words about the role of karst as the source of  water in the hot  and desert areas in North Africa,  Turkey, Crimea, Equatorial America.

Also role of the Karst as the traps for the gold and other noble metals and even diamonds placers.

So  the paper interesting and may be published but   in this case – major revision

Best wishes Igor Ashchepkov

Author Response

Dear Reviewer,

I have accepted all your recommendations and I corrected the manuscript.

Reviewer 2 Report

This is good, well-summarized and clearly presented review of karst types worldwide.

My minor comments are summarized below:

Figure 1 is a little bit difficult to read. Any possibility to present it in color?

Lines 35-36 - could you please elaborate as to what specific karst characteristics will assist classification of karst areas into various karst types. This sentence is somewhat confusing.

Lines 40-41 - what about rock deformations, e.g. mechanical fracturing, faulting, folding. Do these rock properties affect dissolution intensity?

Line 41 - I think some sort of a reference (references) is needed at the end of this sentence.

Line 51 - reference for archeic features?

I assume that Table 1 is based on literature data. Data sources should be provided (referenced) in the Notes to Table 1.

Line 101 - I believe that all references in the text should be numerical in MDPI publications. So [31] (Waltham, Fookes 2003) appears to be duplicated.

Figures 2, 3, 4 and 6 - I assume that all photos were made by the author. If not, sources of photography should be specified in the relevant figure captions.

Author Response

Dear Reviewer,

I corrected most of the comments. However there are 2 exceptions.

Table 1 was not prepared based on literary data thus, no source was indicated, but in the notice I refer to the work which gave a basis to the table.

All photos (except 1) were taken by me. I indicated the source of the exception.

Round 2

Reviewer 1 Report

It seems that paper may be accepted

Reviewer 2 Report

I think that the author did a great job of significantly upgrading the original manuscript. The new version is certainly ready for an immediate publication.